:ᐰ: PLOS ONE

# Volatile anesthetics versus total intravenous anesthesia in patients undergoing coronary artery bypass grafting: An updated meta-analysis and trial sequential analysis of randomized controlled trials

**Xue-feng Jiao**[1,2,3,4], **Xue-mei Lin**[3,5], **Xiao-feng Ni**[1,2,3,4], **Hai-long Li**[1,2,3], **Chuan Zhang**[1,2,3], **Chun-song Yang**[1,2,3], **Hao-xin Song**[1,2,3], **Qiu-sha Yi**[1,2,3,4], **Ling-li Zhang**[1,2,3] *

1 Department of Pharmacy, West China Second University Hospital, Sichuan University, Sichuan, China,
2 Evidence-Based Pharmacy Center, West China Second University Hospital, Sichuan University, Sichuan, China, 3 Key Laboratory of Birth Defects and Related Diseases of Women and Children, Sichuan University, Ministry of Education, Sichuan, China, 4 West China School of Medicine, Sichuan University, Sichuan, China, 5 Department of Anesthesiology, West China Second University Hospital, Sichuan University, Sichuan, China

* zhanglingli@scu.edu.cn

**Data Availability Statement:** All relevant data are within the paper and its Supporting Information files.

## Abstract

### Background

The benefits of volatile anesthetics in coronary artery bypass grafting (CABG) patients remain controversial. We aimed to conduct an updated meta-analysis to assess whether the use of volatile anesthetics during CABG could reduce mortality and other outcomes.

### Methods

We searched eight databases from inception to June 2019 and included randomized controlled trials (RCTs) comparing the effects of volatile anesthetics versus total intravenous anesthesia (TIVA) in CABG patients. The primary outcomes were operative mortality and one-year mortality. The secondary outcomes included the length of stay in the intensive care unit (ICU) and hospital and postoperative safety outcomes (myocardial infarction, heart failure, arrhythmia, stroke, delirium, postoperative cognitive impairment, acute kidney injury, and the use of intra-aortic balloon pump (IABP) or other mechanical circulatory support). Trial sequential analysis (TSA) was performed to control for random errors.

### Results

A total of 89 RCTs comprising 14,387 patients were included. There were no significant differences between the volatile anesthetics and TIVA groups in operative mortality (relative risk (RR) = 0.92, 95% confidence interval (CI): 0.68–1.24, p = 0.59, $I^2$ = 0%), one-year mortality (RR = 0.64, 95% CI: 0.32–1.26, p = 0.19, $I^2$ = 51%), or any of the postoperative safety outcomes. The lengths of stay in the ICU and hospital were shorter in the volatile

**Funding:** This study was supported by Natural Science Foundation of China: Evidence based establishment of evaluation index system for pediatric rational drug use in China (No. 81373381), and National Science and Technology Major Project: Construction of pediatric new drug clinical evaluation technology platform (No. 2017ZX09304029) to LZ. The funders had no role in study design, data collection and analysis, decision to publish, or preparation of the manuscript.

**Competing interests:** The authors have declared that no competing interests exist.

anesthetics group than in the TIVA group. TSA revealed that the results for operative mortality, one-year mortality, length of stay in the ICU, heart failure, stroke, and the use of IABP were inconclusive.

## Conclusions

Conventional meta-analysis suggests that the use of volatile anesthetics during CABG is not associated with reduced risk of mortality or other postoperative safety outcomes when compared with TIVA. TSA shows that the current evidence is insufficient and inconclusive. Thus, future large RCTs are required to clarify this issue.

## Introduction

Coronary artery bypass grafting (CABG) is one of the most common surgeries performed on patients with coronary artery disease [1]. In America alone, nearly 400,000 CABG surgeries are performed every year [2], and the associated mortality is reported to be 2 to 3% [3].

The anesthesia scheme for CABG commonly consists of intravenous anesthetics only, or a combination of volatile anesthetics and intravenous anesthetics. A great deal of experimental and clinical evidence has demonstrated that volatile anesthetics have cardioprotective effects [4,5], even if administered for only a short time [6]. These effects have been attributed to multiple mechanisms, including modulation of gene expression, mitochondrial function, signaling pathways, G-protein-coupled receptors, and potassium channels [7].

However, the benefits of volatile anesthetics in CABG patients is an intensely disputed topic. In 2006, two meta-analyses of randomized controlled trials (RCTs) suggested that volatile anesthetics were not associated with reduced mortality when compared with total intravenous anesthesia (TIVA) in CABG surgery [8,9]. On the other hand, meta-analyses published later have shown a reduced mortality associated with volatile anesthetics in CABG surgeries [10–14]. Finally, both the guideline from the American College of Cardiology/American Heart Association (ACCF/AHA) and the guideline from the European Association for Cardio-Thoracic Surgery (EACTS) have recommended the use of volatile anesthetics in patients undergoing CABG surgery [15,16]. However, several recent, well-conducted, large-scale RCTs did not show reduced mortality by volatile anesthetics [17–19]. These new RCTs have since doubled the number of trial participants, which may lead in the future to reversals in the results of previous meta-analyses.

Thus, in order to evaluate the most recent available evidence, we performed an updated meta-analysis of RCTs to test the hypothesis that the use of volatile anesthetics during CABG does not result in lower mortality than the use of TIVA. We also aimed to examine the effects of volatile anesthetics on lengths of stay in the intensive care unit (ICU), lengths of stay in hospital, and other postoperative safety outcomes. Moreover, we used trial sequential analysis (TSA) to determine whether the currently available evidence was sufficient and conclusive.

## Materials and methods

This meta-analysis was conducted in accordance with the Cochrane Handbook as well as the Preferred Reporting Items for Systematic Reviews and Meta-Analyses (PRISMA) guidelines [20,21] (S2 Table). In addition, we deposited our protocol in protocols.io; the identifier is dx. doi.org/10.17504/protocols.io.7y2hpye.

## Search strategy

We searched Pubmed, Embase (Ovid), the Cochrane Library, and three Chinese databases, the China Knowledge Resource Integrated Database (CNKI), the VIP Database and the Wanfang Database (up to June 2019). We also searched ongoing clinical trial databases, such as "http://clinicaltrials.gov" and "http://www.controlled-trials.com/". The search terms included the following: volatile anesthetics, halothane, sevoflurane, desflurane, isoflurane, enflurane, methoxyflurane, ether, and coronary artery bypass (S1 Table). Moreover, the references of included studies were manually checked for identifying additional relevant studies.

## Inclusion criteria

Inclusion criteria with reference to the PICOS criteria were as follows: (1) *Participants*: patients undergoing general anesthesia for CABG. (2) *Intervention*: an anesthesia plan that included a volatile anesthetic (such as halothane, sevoflurane, desflurane, isoflurane, enflurane, or ether) without restrictions in times and doses of administration. (3) *Comparison*: TIVA (such as propofol, fentanyl, sufentanil, midazolam, thiopental, or etomidate). (4) *Outcomes*: the primary outcomes were operative mortality and one-year mortality. The secondary outcomes included length of stay in ICU, length of stay in hospital, and postoperative safety outcomes (myocardial infarction, heart failure, arrhythmia, stroke, delirium, postoperative cognitive impairment, acute kidney injury, the use of intra-aortic balloon pump (IABP), and the use of other mechanical circulatory support). Eligible studies should report at least one of the above outcomes. According to the Society of Thoracic Surgeons, operative mortality is defined as "① all-cause mortality occurring during the hospitalization in which the operation was performed, even beyond 30 days; and ② all-cause mortality occurring after hospital discharge, but within 30 days of the operation [22]". (5) *Study design*: RCT.

## Exclusion criteria

Exclusion criteria were as follows: (1) CABG combined with valve surgery; (2) epidural anesthesia included in the anesthetic plan; (3) lack of outcome data; (4) full-text not available; (5) study not published in English or Chinese.

## Study selection

Two independent reviewers (XF Jiao and XF Ni) screened the titles and abstracts to determine potentially relevant initial studies. They then evaluated the full-text articles of potentially relevant studies for eligibility. Reviewers resolved disagreements by consensus, and, if necessary, asked for a third reviewer's adjudication.

## Data extraction

Two independent reviewers (XF Jiao and XF Ni) extracted data from each included study using a pre-piloted data extraction form. This included general study information, general patient characteristics, sample sizes, interventions, comparisons, and outcomes. Postoperative safety outcomes occurring during hospitalization or within 30 days of operation were extracted for meta-analysis.

## Risk of bias in individual studies

Two independent reviewers (XF Jiao and XF Ni) evaluated the risk of bias in each selected study using the Cochrane risk of bias tool. This tool consists of seven items: random sequence generation, allocation concealment, blinding of participants and personnel, blinding of

outcome assessment, incomplete outcome data, selective reporting, and other biases. According to the Cochrane Handbook for Systematic Reviews of Interventions, we assessed the risk of bias per outcome across trials as: low risk of bias (if the seven items were all evaluated as at low risk of bias); unclear risk of bias (if one or more items were evaluated as at unclear risk of bias); high risk of bias (if one or more items were evaluated as at high risk of bias) [23].

## Quality of evidence

We used the Grading of Recommendations Assessment, Development, and Evaluation (GRADE) approach to assess the overall level of evidence strength of each outcome [24]. By considering five categories of limitations (risk of bias, inconsistencies, indirectness, imprecisions, and reporting biases), the GRADE approach provided a rating of quality of evidence (high, moderate, low, or very low) for each outcome [25].

## Data synthesis and analysis

For dichotomous data, we extracted the numbers of events and total numbers for each group and adopted the risk ratio (RR) with 95% confidence interval (CI) as the effect measure. For continuous data, we extracted mean and standard deviation (SD), and adopted mean difference (MD) with 95% confidence interval (CI) as the effect measure. If the included studies only provided the median, range and/or the first and third quartiles, we first transformed these data to sample means and SDs through published formulas [26]. If there were more than one volatile anesthetics or TIVA groups in the included studies, these were combined for the pooled analyses. I-squared ($I^2$) test was used to assess heterogeneity. If heterogeneity was acceptable ($I^2 \leq 50\%$), a fixed effect model was used. If heterogeneity was significant ($I^2 > 50\%$), a random effects model was used and sensitivity analysis was conducted by excluding one or more studies at a time to assess stability of results. Moreover, if ten or more studies were included, a funnel plot was used to assess publication bias. Statistical analyses were conducted by RevMan 5.3.

For all primary and secondary outcomes, we conducted TSA to reduce the risk of random errors. TSA is a method which calculates the required information size (RIS) for meta-analysis, constructing both the trial sequential monitoring boundaries for benefit or harm, and the futility boundary before reaching RIS. The risk of type I error was set at 5% with a power of 80%. For dichotomous outcomes, we initially calculated RIS to detect a 20% relative risk reduction (RRR), as this value is believed to represent a reasonable intervention effect in most therapeutic protocols [27]. If the boundary RIS was ignored due to too little information use, we then tested for a 30% RRR. We used the median event proportion of TIVA groups (excluding zero-event trials) as the control group event proportion. For continuous outcomes, we calculated RIS based on empirical estimation from TSA software [27]. TSA was conducted with the TSA viewer version 0.9.5.10 Beta (www.ctu.dk/tsa).

## Results

### Search results

A total of 4289 records were identified by the initial search. After selection, 89 RCTs that met our inclusion criteria were included in this meta-analysis (Fig 1).

### Characteristics of the included studies

The 89 included RCTs comprised 14,387 patients, including 7,719 patients anesthetized with volatile anesthetics and 6,668 patients with TIVA. These RCTs took place in a variety of

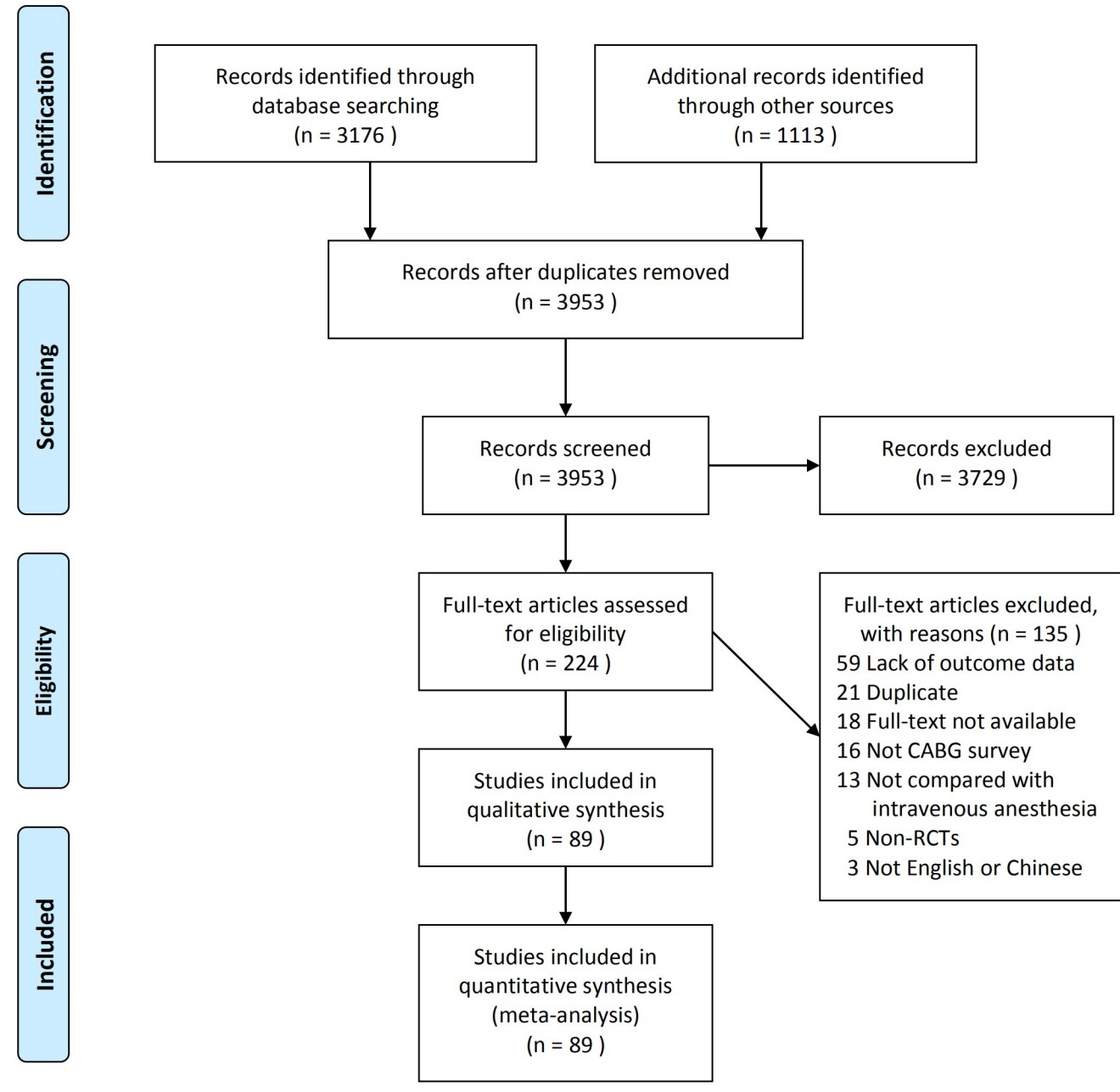

**Fig 1. Flow diagram of study selection (PRISMA format).**

settings: 40 RCTs in Asia, 32 RCTs in Europe, 11 RCTs in North America, 4 RCTs in Australia, 1 RCT in South America, and 1 RCT in Africa. Most RCTs were performed on patients undergoing on-pump CABG surgery, while 27 RCTs were performed on patients undergoing off-pump CABG surgery. Among the included RCTs, 14 were three-arm studies and 3 were four-arm studies. The volatile anesthetic used in the included RCTs varied: 39 RCTs used sevoflurane, 23 RCTs used isoflurane, 8 RCTs used desflurane, 6 RCTs used enflurane, and 13 RCTs used more than one volatile anesthetics. Detailed characteristics of the included RCTs are shown in S4 Table.

### Risk of bias in individual studies

The risk of bias assessment of the included RCTs is shown in S1 Fig. We determined the following features to be at high risk of bias: random sequence generation in 1 RCT; blinding of participants and personnel in 8 RCTs; blinding of outcome assessment in 2 RCTs; incomplete outcome data in 1 RCT; other biases in 2 RCTs. Overall, 5 (6%) RCTs were rated at low risk of bias, 12 (13%) RCTs were rated at high risk of bias, and the remaining RCTs were rated at unclear risk of bias.

### Primary outcomes

**Operative mortality.** A total of 44 RCTs reported operative mortality. There was no statistically significant difference between the volatile anesthetics and TIVA groups in operative mortality (RR = 0.92, 95% CI: 0.68–1.24, p = 0.59), and no heterogeneity ($I^2$ = 0%) (Fig 2).

TSA showed that the accrued information size (n = 11,567) was 35% of RIS (n = 33,174). The cumulative Z-curve crossed neither the trial sequential monitoring boundary nor the futility boundary, indicating that current evidence was insufficient and inconclusive (Fig 3).

The quality of evidence was rated as moderate for this outcome (S5 Table).

**One-year mortality.** Five RCTs reported one-year mortality. There was no statistically significant difference between the volatile anesthetics and TIVA groups in one-year mortality (RR = 0.64, 95% CI: 0.32–1.26, p = 0.19), with moderate heterogeneity ($I^2$ = 51%) (Fig 4). Sensitivity analysis did not change the result for one-year mortality (RR = 0.92, 95% CI: 0.68–1.25, p = 0.61, $I^2$ = 0%) (Table 1).

TSA showed that the accrued information size (n = 5,953) was 27% of RIS (n = 22,248). The cumulative Z-curve crossed neither the trial sequential monitoring boundary nor the futility boundary, indicating that current evidence was insufficient and inconclusive (Fig 5).

The quality of evidence was rated as low for this outcome (S5 Table).

### Secondary outcomes

**Length of stay in ICU.** A total of 43 RCTs reported length of stay in ICU, which was shorter in the volatile anesthetics group than in the TIVA group (MD = -4.14 h, 95% CI: -5.63– -2.66 h, p<0.00001), with high heterogeneity ($I^2$ = 98%) (Table 2). After sensitivity analysis, there was no statistically significant difference between the volatile anesthetics and TIVA groups in length of stay in ICU (MD = -0.01 h, 95% CI: -0.04–0.01 h, p = 0.38), with mild heterogeneity ($I^2$ = 38%) (Table 1).

TSA showed that the accrued information size (n = 8,980) was 13% of RIS (n = 68,085). The cumulative Z-curve crossed neither the trial sequential monitoring boundary nor the futility boundary, indicating that current evidence was insufficient and inconclusive (S2 Fig).

The quality of evidence was rated as very low for this outcome (S5 Table).

### Length of stay in hospital

A total of 34 RCTs reported length of stay in hospital, which was shorter in the volatile anesthetics than in the TIVA group (MD = -1.22 d, 95% CI: -1.81– -0.62 d, p <0.0001), with high heterogeneity ($I^2$ = 95%) (Table 2). After sensitivity analysis, the length of hospital stay in the volatile anesthetics group was shorter than in the TIVA group (MD = -0.16 d, 95% CI: -0.28– -0.04 d, p = 0.008), with mild heterogeneity ($I^2$ = 26%) (Table 1).

TSA showed that the accrued information size (n = 9,267) exceeded RIS (n = 3,945), indicating that the current evidence was sufficient to reach a firm conclusion (S3 Fig).

The quality of evidence was rated as low for this outcome (S5 Table).

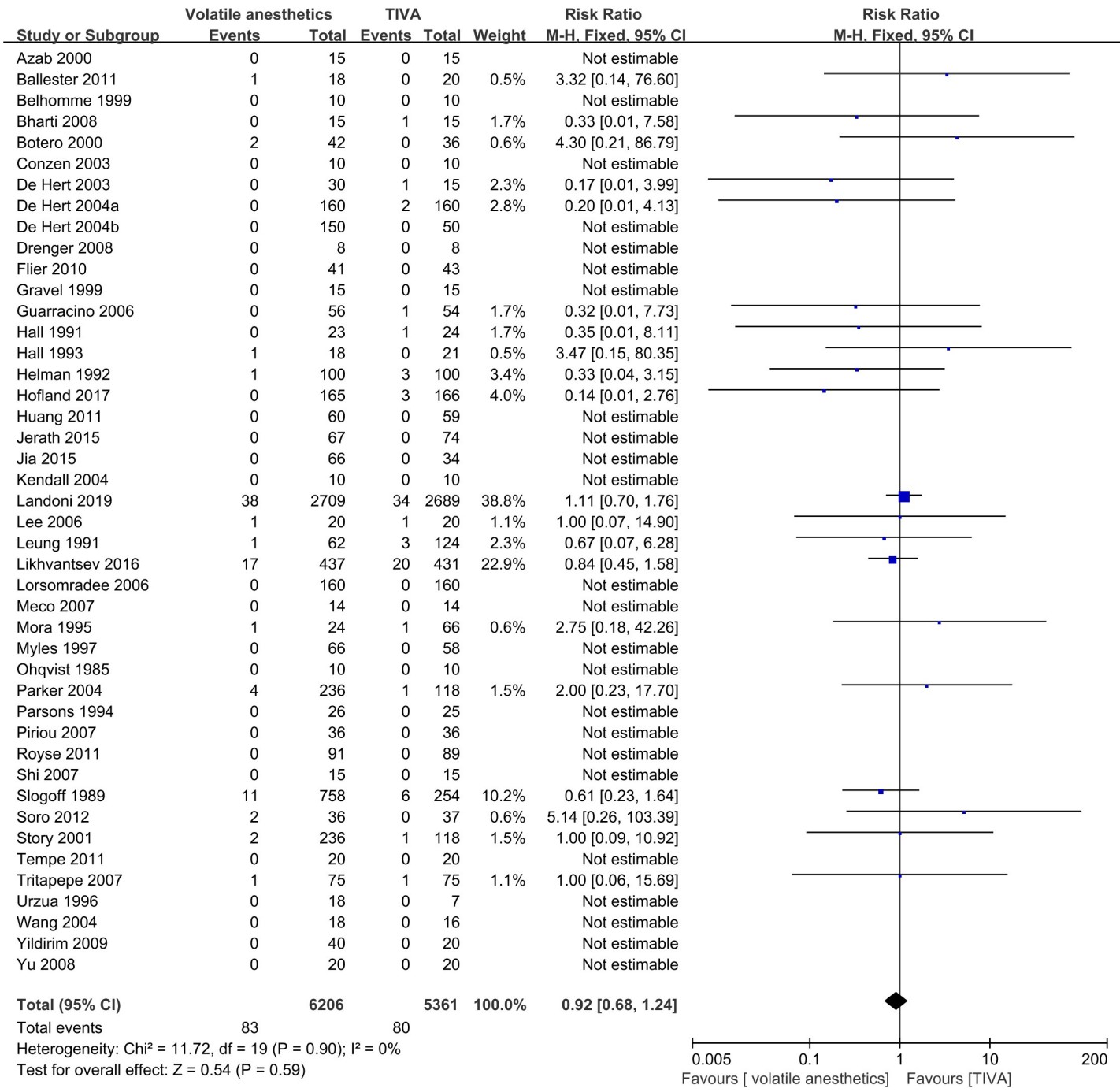

**Fig 2. Forest plot of operative mortality.**

## Myocardial infarction

A total of 28 RCTs reported myocardial infarctions. There was no statistically significant difference between the volatile anesthetics and TIVA groups in the incidence of myocardial infarctions (RR = 0.94, 95% CI: 0.73–1.21, p = 0.64), without heterogeneity ($I^2$ = 0%) (Table 2).

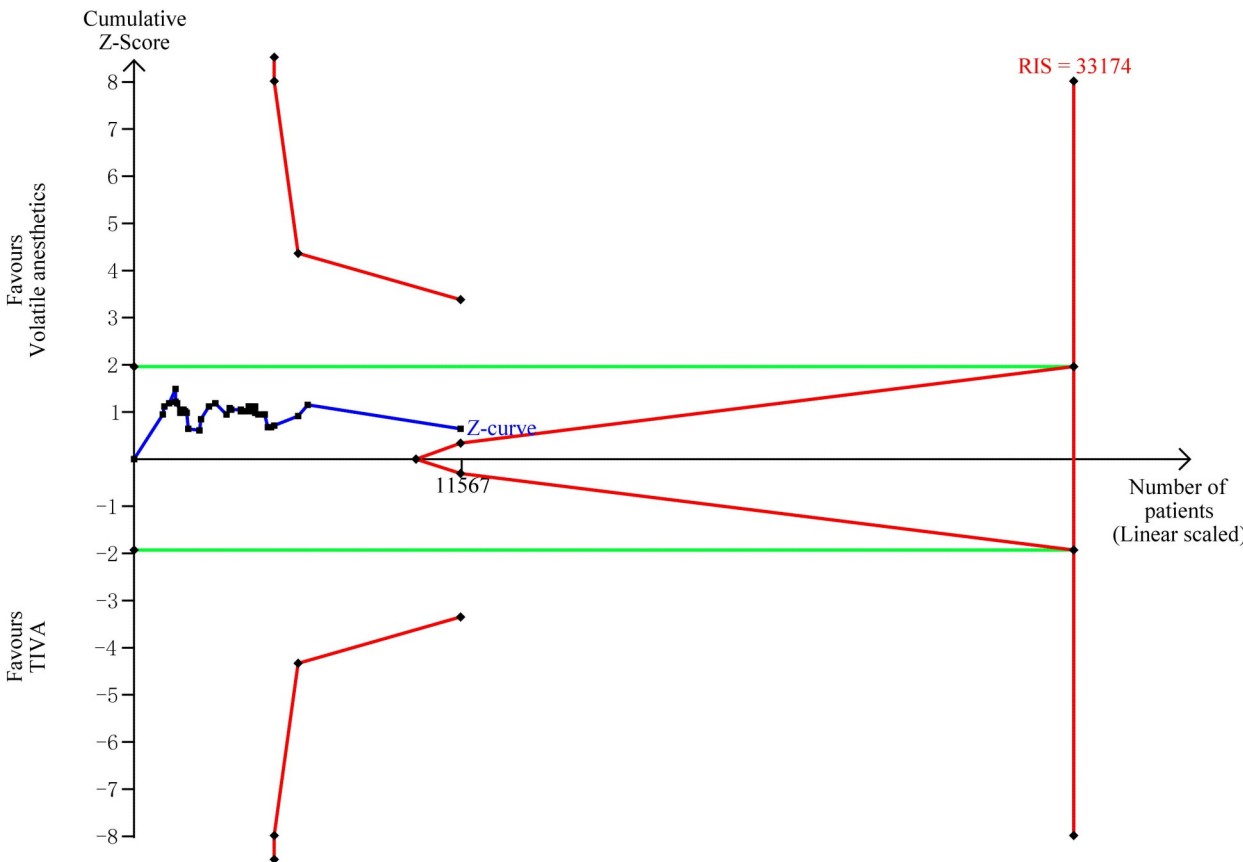

**Fig 3. Trial sequential analysis of operative mortality.** The risk of type I error was set at 5% with a power of 80%. The variance was calculated from the data obtained from the included trials. The relative risk reduction (RRR) was set at 20%.

TSA showed that the accrued information size (n = 8,574) was 44% of RIS (n = 19,590). The cumulative Z-curve crossed the futility boundary, indicating that current evidence was sufficient to reach a firm conclusion (S4 Fig).

The quality of evidence was rated as high for this outcome (S5 Table).

## Heart failure

A total of 4 RCTs reported heart failures. There was no statistically significant difference between the volatile anesthetics and TIVA groups in the incidence of heart failures (RR = 0.39, 95% CI: 0.08–2.01, p = 0.26), without heterogeneity ($I^2$ = 0%) (Table 2).

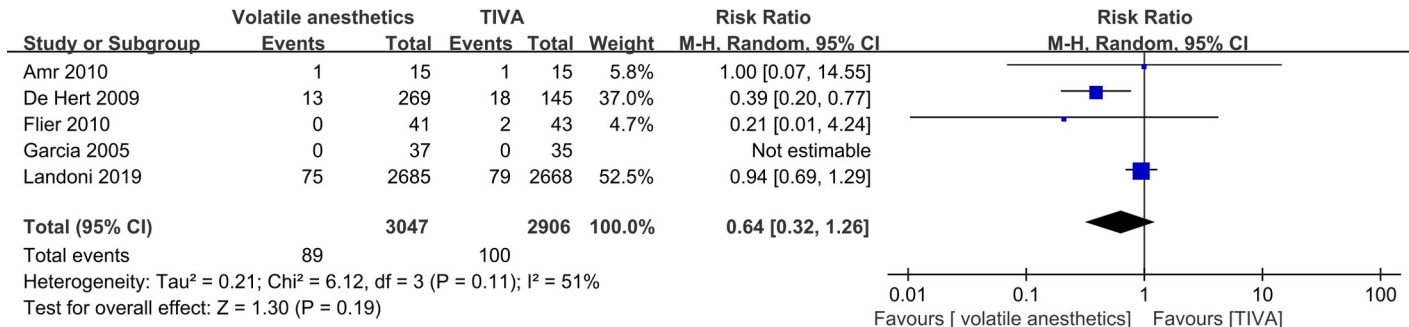

**Fig 4. Forest plot of one-year mortality.**

**Table 1. Sensitivity analysis.**

| Outcome | Number of RCTs | Number of patients | MD or RR | 95% CI | *p*-value | $I^2$ | Model |
|---|---|---|---|---|---|---|---|
| One-year mortality | 4 | 5539 | 0.92 | 0.68–1.25 | 0.61 | 0% | Fix |
| Length of stay in ICU | 32 | 7345 | -0.01 h | -0.04–0.01 h | 0.38 | 36% | Fix |
| Length of stay in hospital | 29 | 7535 | -0.16 d | -0.28– -0.04 d | 0.008 | 26% | Fix |
| Postoperative cognitive impairment | 6 | 5565 | 1.37 | 0.94–1.98 | 0.1 | 36% | Fix |

RCTs, randomized controlled trials; MD, mean difference; RR, relative risk; CI, confidence interval; $I^2$, statistical heterogeneity; ICU, intensive care unit; h, hour; d, day.

TSA showed that the accrued information size (n = 461) was 5% of RIS (n = 8,908). The cumulative Z-curve crossed neither the trial sequential monitoring boundary nor the futility boundary, indicating that current evidence was insufficient and inconclusive (S5 Fig).

The quality of evidence was rated as moderate for this outcome (S5 Table).

## Arrhythmia

A total of 29 RCTs reported arrhythmias. There was no statistically significant difference between the volatile anesthetics and TIVA groups in the incidence of arrhythmias (RR = 0.89, 95% CI: 0.77–1.03, p = 0.11), without heterogeneity ($I^2$ = 0%) (Table 2).

TSA showed that the accrued information size (n = 3,084) exceeded RIS (n = 2,897), indicating that the current evidence was sufficient to reach a firm conclusion (S6 Fig).

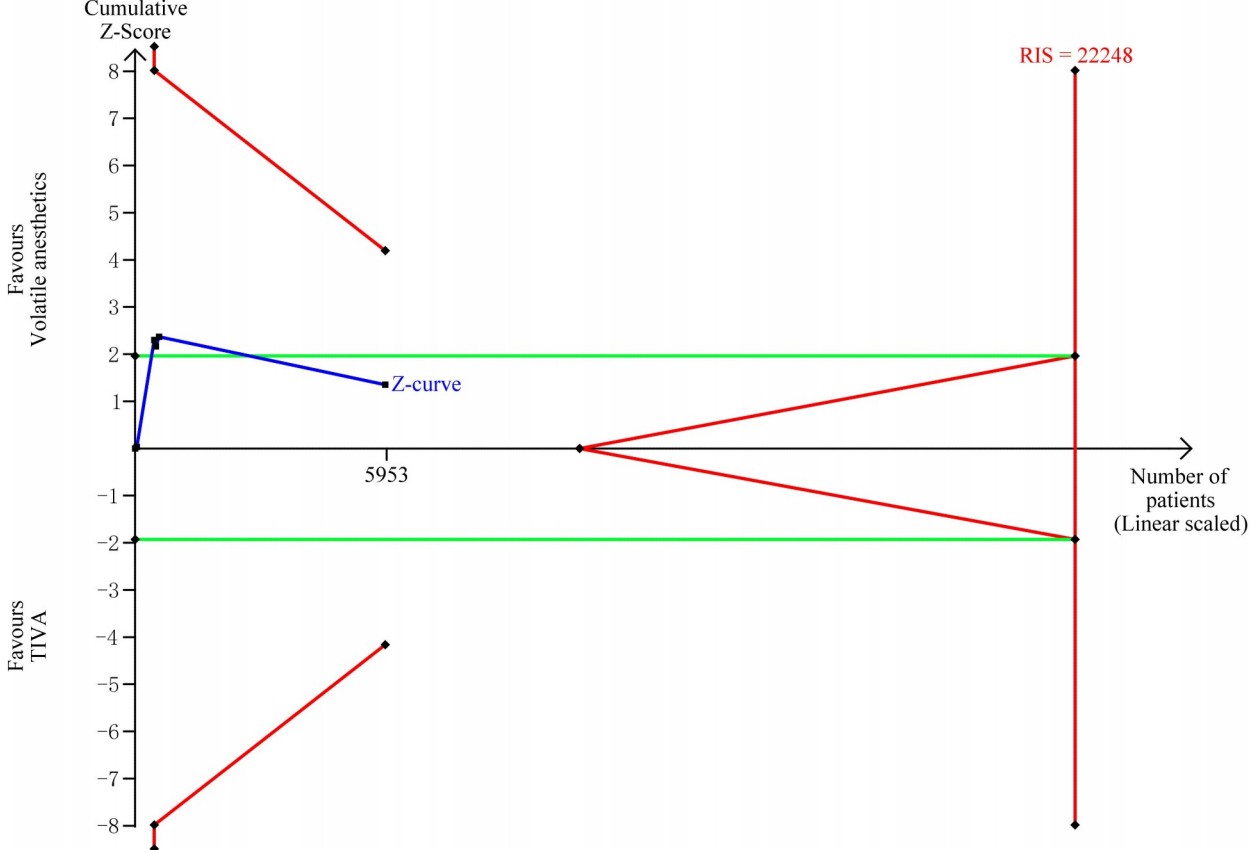

**Fig 5. Trial sequential analysis of one-year mortality.** The risk of type I error was set at 5% with a power of 80%. The variance was calculated from the data obtained from the included trials. The relative risk reduction (RRR) was set at 20%.

**Table 2. Results of meta-analysis for secondary outcomes.**

| Outcome | Number of RCTs | Number of patients | MD or RR | 95% CI | *p*-value | I² | Model |
|---|---|---|---|---|---|---|---|
| Length of stay in ICU | 43 | 8980 | -4.14 h | -5.63– -2.66 h | <0.00001 | 98% | Random |
| Length of stay in hospital | 34 | 9267 | -1.22 d | -1.81– -0.62 d | <0.0001 | 95% | Random |
| **Postoperative safety outcomes** | | | | | | | |
| Myocardial infarction | 28 | 8574 | 0.94 | 0.73–1.21 | 0.64 | 0% | Fix |
| Heart failure | 4 | 461 | 0.39 | 0.08–2.01 | 0.26 | 0% | Fix |
| Arrhythmia | 29 | 3084 | 0.89 | 0.77–1.03 | 0.11 | 0% | Fix |
| Stroke | 2 | 5353 | 1.46 | 0.76–2.81 | 0.25 | 0% | Fix |
| Delirium | 3 | 5582 | 0.96 | 0.71–1.29 | 0.78 | 0% | Fix |
| Postoperative cognitive impairment | 8 | 5792 | 1.20 | 0.74–1.94 | 0.46 | 59% | Random |
| Acute kidney injury | 5 | 5706 | 0.98 | 0.79–1.22 | 0.88 | 0% | Fix |
| The use of IABP | 6 | 5587 | 0.67 | 0.29–1.54 | 0.34 | 0% | Fix |
| The use of other mechanical circulatory support | 4 | 5458 | 0.73 | 0.33–1.63 | 0.45 | 40% | Fix |

RCTs, randomized controlled trials; MD, mean difference; RR, relative risk; CI, confidence interval; I², statistical heterogeneity; ICU, intensive care unit; h, hour; d, day; IABP, intra-aortic balloon pump.

The quality of evidence was rated as high for this outcome (S5 Table).

## Stroke

A total of 2 RCTs reported strokes. There was no statistically significant difference between the volatile anesthetics and TIVA groups in the incidence of strokes (RR = 1.46, 95% CI: 0.76–2.81, p = 0.25), without heterogeneity (I² = 0%) (Table 2).

TSA showed that the accrued information size (n = 5,353) was 10% of RIS (n = 52,433). The cumulative Z-curve crossed neither the trial sequential monitoring boundary nor the futility boundary, indicating that the current evidence was insufficient and inconclusive (S7 Fig).

The quality of evidence was rated as moderate for this outcome (S5 Table).

## Delirium

A total of 3 RCTs reported delirium. There was no statistically significant difference between the volatile anesthetics and TIVA groups in the incidence of delirium (RR = 0.96, 95% CI: 0.71–1.29, p = 0.78), without heterogeneity (I² = 0%) (Table 2).

TSA showed that the accrued information size (n = 5,582) exceeded RIS (n = 3,286), indicating that the current evidence was sufficient to reach a firm conclusion (S8 Fig).

The quality of evidence was rated as moderate for this outcome (S5 Table).

## Postoperative cognitive impairment

A total of 8 RCTs reported postoperative cognitive impairment. There was no statistically significant difference between the volatile anesthetics and TIVA groups in the incidence of postoperative cognitive impairment (RR = 1.20, 95% CI: 0.74–1.94, p = 0.46), with moderate heterogeneity (I² = 59%) (Table 2). Sensitivity analysis did not change the result for postoperative cognitive impairment (RR = 1.37, 95% CI: 0.94–1.98, p = 0.1, I² = 36%) (Table 1).

TSA showed that the accrued information size (n = 5,792) was 52% of RIS (n = 11,082). The cumulative Z-curve crossed the futility boundary, indicating that the current evidence was sufficient to reach a firm conclusion (S9 Fig).

The quality of evidence was rated as low for this outcome (S5 Table).

### Acute kidney injury

A total of 5 RCTs reported acute kidney injury. There was no statistically significant difference between the volatile anesthetics and TIVA groups in the incidence of acute kidney injury (RR = 0.98, 95% CI: 0.79–1.22, p = 0.88), without heterogeneity ($I^2$ = 0%) (Table 2).

TSA showed that the accrued information size (n = 5,706) was 55% of RIS (n = 10,385). The cumulative Z-curve crossed the futility boundary, indicating that the current evidence was sufficient to reach a firm conclusion (S10 Fig).

The quality of evidence was rated as high for this outcome (S5 Table).

### The use of IABP

A total of 6 RCTs reported the use of IABPs. There was no statistically significant difference between the volatile anesthetics and TIVA groups in the use of IABPs (RR = 0.67, 95% CI: 0.29–1.54, p = 0.34), without heterogeneity ($I^2$ = 0%) (Table 2).

TSA showed that the accrued information size (n = 5,587) was 41% of RIS (n = 13,493). The cumulative Z-curve crossed neither the trial sequential monitoring boundary nor the futility boundary, indicating that the current evidence was insufficient and inconclusive (S11 Fig).

The quality of evidence was rated as moderate for this outcome (S5 Table).

### The use of other mechanical circulatory support

A total of 4 RCTs reported the use of other mechanical circulatory support. There was no statistically significant difference between the volatile anesthetics and TIVA groups in the use of other mechanical circulatory support (RR = 0.73, 95% CI: 0.33–1.63, p = 0.45), with mild heterogeneity ($I^2$ = 40%) (Table 2).

TSA showed that the accrued information size (n = 5,458) was 64% of RIS (n = 8,590). The cumulative Z-curve crossed the futility boundary, indicating that the current evidence was sufficient to reach a firm conclusion (S12 Fig).

The quality of evidence was rated as moderate for this outcome (S5 Table).

### Publication bias

No significant publication bias was found with respect to operative mortality (Fig 6).

## Discussion

### Main findings

To our knowledge, this is the most comprehensive meta-analysis comparing the effects of volatile anesthetics versus TIVA in patients undergoing CABG surgery and is the first study to investigate these effects using the TSA method. Our conventional meta-analysis suggested that there were no statistically significant differences in operative mortality, one-year mortality, myocardial infarction, heart failure, arrhythmia, stroke, delirium, postoperative cognitive impairment, acute kidney injury, the use of IABP, and the use of other mechanical circulatory support between the volatile anesthetics and TIVA groups after CABG surgery. However, volatile anesthetics were associated with reductions in length of stay in ICU and hospital when compared with TIVA. TSA showed that the results for length of stay in hospital, myocardial infarction, arrhythmia, delirium, postoperative cognitive impairment, acute kidney injury, and the use of other mechanical circulatory support were conclusive, while the results for operative mortality, one-year mortality, length of stay in ICU, heart failure, stroke, and the use of IABP were inconclusive.

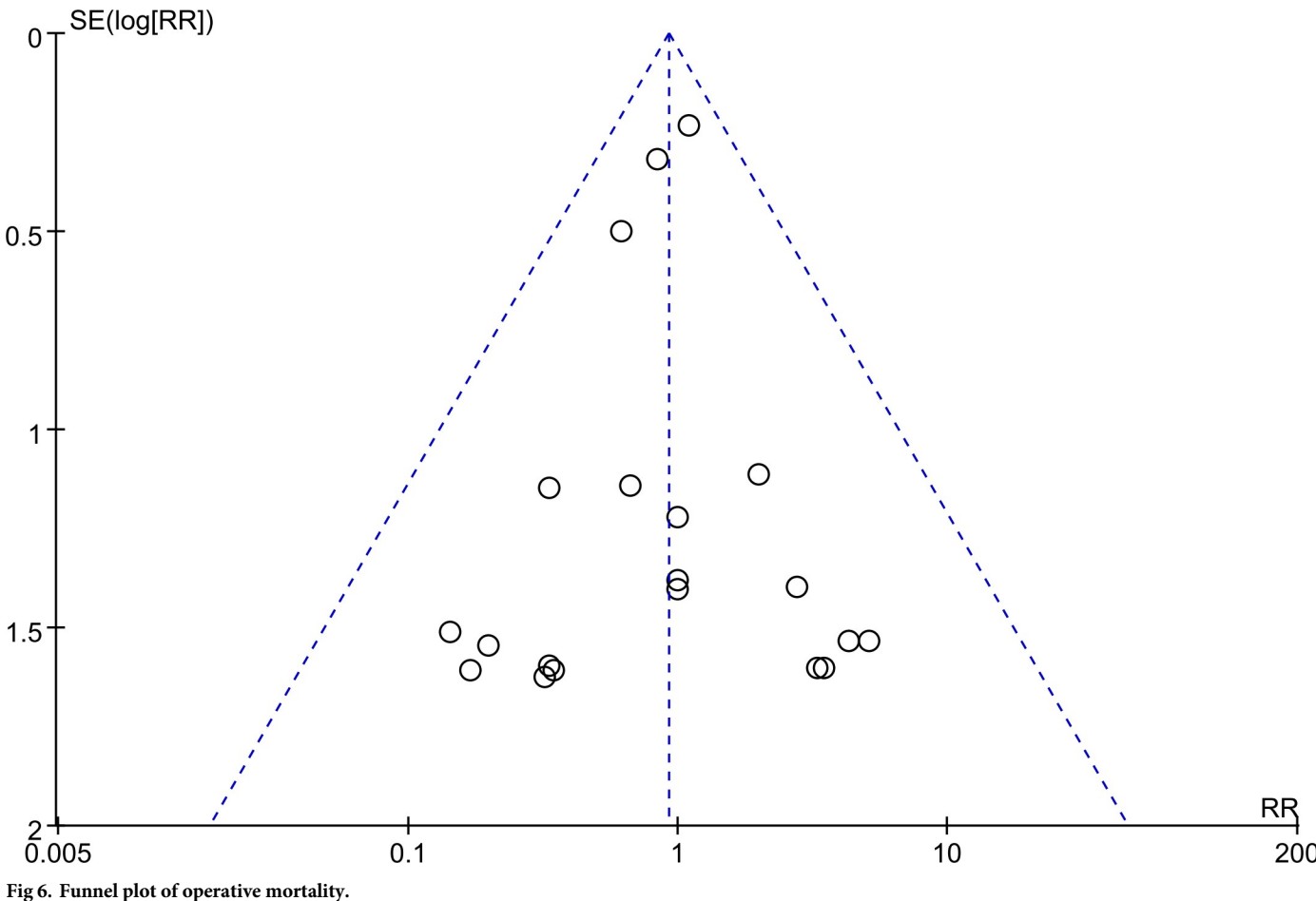

**Fig 6. Funnel plot of operative mortality.**

## Compared with previous meta-analyses and RCTs

Our meta-analysis did not reveal reductions in mortality by the use of volatile anesthetics in CABG surgery, a result which was similar to those of meta-analyses published in 2006. However, the meta-analyses published after 2006 were not consistent with our results. For example, the meta-analyses of EI Dib et al. (2017), Uhlig et al. (2016), Zangrillo et al. (2015), Landoni et al. (2013) and Landoni et al. (2007) all showed that volatile anesthetics could reduce mortality when compared with TIVA in CABG survey [10–14]. The meta-analyses of EI Dib et al. (2017) and Landoni et al. (2007) also indicated that volatile anesthetics could reduce the risk of myocardial infarction when compared with TIVA in CABG surgery [10,14]. Moreover, the meta-analyses of Cai et al. (2014) indicated that volatile anesthetics could reduce the incidence of acute kidney injury when compared with TIVA in CABG survey [28]. These inconsistencies may be due to differences in the numbers of included RCTs and patients. In previous meta-analyses, the largest sample size was 58 RCTs with 6105 patients [10], while the smallest sample size was only 10 RCTs with 1600 patients [28]. However, our meta-analysis included 89 RCTs with 14,387 patients, much larger numbers than those of any previous meta-analyses. Moreover, we further performed TSA to assess the adequacy of current evidence and the likelihood of future RCTs altering the present conclusions. Thus, our meta-analysis provides more robust and precise estimations of these outcomes.

Recently Landoni et al. published a well-designed, multicenter RCT encompassing 5400 patients, which was much larger than any of the previous RCTs on this topic [17]. This largest RCT failed to reveal any benefits of volatile anesthetics in CABG patients, which was quite contrary to the results of several previous meta-analyses. However, whether this RCT had enough statistical power to counter the results of previous meta-analyses and end further discussions about the benefit of volatile anesthetics was not known. Thus, we performed an updated and comprehensive meta-analysis on this topic. After pooling all the available evidence, the majority of our results were similar to the results of Landoni et al.; however, the results for length of stay in ICU and hospital were not consistent with the results of Landoni et al. Furthermore, our TSA results indicated that current evidence for mortality as well as several other postoperative safety outcomes was insufficient and inconclusive, and more RCTs would be required to further clarify this issue. Thus, although of large sample size, the Landoni et al. RCT is not convincing enough to end any further discussion on this topic.

## Heterogeneity

In our meta-analysis, the heterogeneities associated with length of stay in ICU and hospital were relatively high. One possible reason was that some of the included RCTs provided the median, range, and/or the first and third quartiles for these outcomes. Transformation of these data to sample means and SDs for meta-analysis may have introduced errors, thus causing heterogeneity between studies [26]. Other reasons may include variations in ethnicity, age, severity of disease, treatment regimens, operation types and anesthesia schemes. It is noteworthy that the heterogeneities may also be attributed to variations in settings, hospitals, medical insurance systems, economy levels and cultural differences. However, these were not analyzed in our meta-analysis due to insufficient data. We used a random effects model to account for these heterogeneities and found shorter lengths of both ICU stay and hospital stay in the volatile anesthetics than in the TIVA group. Furthermore, we conducted sensitivity analysis to reduce these heterogeneities, after which the result for length of stay in ICU was not statistically reliable. In addition, even though the result for length of stay in hospital was statistically reliable after sensitivity analysis, it amounted to only a 0.16-day reduction by volatile anesthetics, which was of little clinical value. Thus, the results concerning length of stay in ICU and hospital should be interpreted cautiously.

## Quality of evidence

According to GRADE, only three outcomes (myocardial infarction, arrhythmia, and acute kidney injury) in our meta-analysis were rated as of high quality. Of the other outcomes, six (operative mortality, heart failure, stroke, delirium, the use of IABP, and the use of other mechanical circulatory support) were rated down to moderate quality, three (one-year mortality, length of stay in hospital, and postoperative cognitive impairment) were rated down to low quality, and one (length of stay in ICU) was rated down to very low quality. The reasons for rating down the quality of evidence for the above outcomes included imprecisions revealed by TSA, significant heterogeneity, lack of or unclear blinding, and large differences between point estimates. Overall, in accordance with the primary outcomes, we have low confidence that the use of volatile anesthetics is superior to TIVA for patients undergoing CABG surgery [29].

## Implications for practice

Nowadays, volatile anesthetics have seen widespread use in CABG surgery. Particularly in America, nearly all patients undergoing CABG surgery receive an anesthesia plan containing

volatile anesthetics [15]. Furthermore, both the ACCF/AHA and the EACTS guidelines give a class IIa or class I recommendation for the use of volatile anesthetics in CABG patients [15,16]. However, our results suggest that although the current evidence is insufficient and inconclusive, the use of volatile anesthetics may not be superior to TIVA in CABG patients. Thus, based on our results, the widespread use of volatile anesthetics in CABG surgery and the current guidelines regarding this topic may be not appropriate.

## Strengths and limitations

The strengths of our meta-analysis include the large sample size, the comprehensive outcomes, the use of multiple sensitivity analyses, the assessment of reliability of current evidence by TSA, and the assessment of quality of evidence for each outcome. On the other hand, our meta-analysis also has certain limitations. First, we only focused on patients who underwent CABG surgery. Thus, our results cannot be extrapolated to patients undergoing other types of cardiac surgery. Second, the majority of the included RCTs had an unclear risk of bias, with only five (6%) RCTs having a low risk of bias. Third, we only included RCTs published in English or Chinese, which might result in a language bias. Fourth, our meta-analysis was heavily influenced by the Landoni et al. RCT [15]. For example, with respect to primary outcomes, this large RCT contributed a weight of evidence of 38.8% for operative mortality and 52.5% for one-year mortality, which were quite high compared to the number of included studies.

## Conclusions

Conventional meta-analysis suggests that the use of volatile anesthetics during CABG is not associated with reduced risks of operative mortality, one-year mortality, and postoperative safety outcomes (myocardial infarction, heart failure, arrhythmia, stroke, delirium, postoperative cognitive impairment, acute kidney injury, the use of IABP and the use of other mechanical circulatory support) when compared with TIVA. TSA shows that the current evidence for mortality as well as several other postoperative safety outcomes (heart failure, stroke, and the use of IABP) is insufficient and inconclusive. Thus, the use of volatile anesthetics may not be superior to TIVA for CABG patients. Future large RCTs are still needed to clarify this issue.

## Supporting information

**S1 Table. Search strategy in PubMed.**
(DOCX)

**S2 Table. PRISMA 2009 checklist.**
(DOCX)

**S3 Table. Reference list of included RCTs.**
(DOCX)

**S4 Table. Characteristics of the included studies.**
(DOCX)

**S5 Table. Grade evidence profile for each outcome.**
(DOCX)

**S1 Fig. Risk of bias for each included RCT.**
(TIF)

**S2 Fig. Trial sequential analysis of length of stay in intensive care unit (ICU).** The risk of type I error was set at 5% with a power of 80%. The variance was calculated from the data obtained from the included trials. The mean difference reduction was set at -0.05 h.
(TIF)

**S3 Fig. Trial sequential analysis of length of stay in hospital.** The risk of type I error was set at 5% with a power of 80%. The variance was calculated from the data obtained from the included trials. The relative risk reduction (RRR) was set at 20%.
(TIF)

**S4 Fig. Trial sequential analysis of myocardial infarction.** The risk of type I error was set at 5% with a power of 80%. The variance was calculated from the data obtained from the included trials. The mean difference reduction was set at -1.18 d.
(TIF)

**S5 Fig. Trial sequential analysis of heart failure.** The risk of type I error was set at 5% with a power of 80%. The variance was calculated from the data obtained from the included trials. The relative risk reduction (RRR) was set at 30%.
(TIF)

**S6 Fig. Trial sequential analysis of arrhythmia.** The risk of type I error was set at 5% with a power of 80%. The variance was calculated from the data obtained from the included trials. The relative risk reduction (RRR) was set at 20%.
(TIF)

**S7 Fig. Trial sequential analysis of stroke.** The risk of type I error was set at 5% with a power of 80%. The variance was calculated from the data obtained from the included trials. The relative risk reduction (RRR) was set at 30%.
(TIF)

**S8 Fig. Trial sequential analysis of delirium.** The risk of type I error was set at 5% with a power of 80%. The variance was calculated from the data obtained from the included trials. The relative risk reduction (RRR) was set at 20%.
(TIF)

**S9 Fig. Trial sequential analysis of postoperative cognitive impairment.** The risk of type I error was set at 5% with a power of 80%. The variance was calculated from the data obtained from the included trials. The relative risk reduction (RRR) was set at 20%.
(TIF)

**S10 Fig. Trial sequential analysis of acute kidney injury.** The risk of type I error was set at 5% with a power of 80%. The variance was calculated from the data obtained from the included trials. The relative risk reduction (RRR) was set at 20%.
(TIF)

**S11 Fig. Trial sequential analysis of the use of intra-aortic balloon pump (IABP).** The risk of type I error was set at 5% with a power of 80%. The variance was calculated from the data obtained from the included trials. The relative risk reduction (RRR) was set at 20%.
(TIF)

**S12 Fig. Trial sequential analysis of the use of other mechanical circulatory support.** The risk of type I error was set at 5% with a power of 80%. The variance was calculated from the data obtained from the included trials. The relative risk reduction (RRR) was set at 20%.
(TIF)

## Acknowledgments

We thank Group of People with Highest Risk of Drug Exposure of International Network for the Rational Use of Drugs, China and Evidence-based Pharmacy Committee of Chinese Pharmaceutical Association for helping to coordinate the authors to review earlier manuscript drafts and provide valuable comments.

## Author Contributions

**Conceptualization:** Xue-feng Jiao, Ling-li Zhang.

**Data curation:** Xue-feng Jiao, Xiao-feng Ni, Hai-long Li.

**Formal analysis:** Xue-feng Jiao, Xiao-feng Ni, Chun-song Yang.

**Funding acquisition:** Ling-li Zhang.

**Investigation:** Xue-feng Jiao, Xue-mei Lin, Xiao-feng Ni, Chuan Zhang, Hao-xin Song.

**Methodology:** Xue-feng Jiao, Xiao-feng Ni, Hai-long Li, Chun-song Yang.

**Project administration:** Xue-feng Jiao, Qiu-sha Yi.

**Resources:** Ling-li Zhang.

**Software:** Xiao-feng Ni, Hao-xin Song.

**Supervision:** Xue-mei Lin, Ling-li Zhang.

**Validation:** Xue-feng Jiao, Chuan Zhang.

**Writing – original draft:** Xue-feng Jiao.

**Writing – review & editing:** Xue-mei Lin, Hai-long Li, Ling-li Zhang.

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
