## [Decision Letter · Decision Letter 0]

16 Sep 2019

PONE-D-19-23473

Volatile anesthetics versus total intravenous anesthesia for patients undergoing coronary artery bypass grafting: An updated meta-analysis of 90 randomized controlled trials

PLOS ONE

Dear Dr Zhang,

Thank you for submitting your manuscript to PLOS ONE. After careful consideration, we feel that it has merit but does not fully meet PLOS ONE’s publication criteria as it currently stands. Therefore, we invite you to submit a revised version of the manuscript that addresses the points raised during the review process.

This manuscript is an appropriately and well written structured meta-analysis. The topic is interesting and has clinical relevance. Numerous previous meta-analyses were conducted on the topic so the findings are not surprising. The multicenter large randomized clinical trial by Giovanni Landoni and colleagues recently published on NEJM, stopped for futility after 50% of the proposed patients were enrolled, should stop any other discussion on this specific topic. Evidence provided by a randomized study is certainly stronger then the results of a meta-analysis, and the findings of the current meta-analysis are influenced by the results of Landoni et al.

Please put in evidence on your limitations these comments, and answer to the question on the review section. 

The grammar should be correct by an english native speaker. 

In the manuscript N2O is included among the "volatiles". I think this is a procedural mistake. I suggest you to change the title or alternatively remove these studies.   

There are not conflicts between the reviews. 

We would appreciate receiving your revised manuscript by October 6 th.

To enhance the reproducibility of your results, we recommend that if applicable you deposit your laboratory protocols in protocols.io, where a protocol can be assigned its own identifier (DOI) such that it can be cited independently in the future. For instructions see: http://journals.plos.org/plosone/s/submission-guidelines#loc-laboratory-protocols

We look forward to receiving your revised manuscript.

Kind regards,

Martina Crivellari

Academic Editor

PLOS ONE

Journal Requirements:

Reviewers' comments:

Reviewer's Responses to Questions

**Comments to the Author**

1. Is the manuscript technically sound, and do the data support the conclusions?

Reviewer #1: Yes

Reviewer #2: Yes

2. Has the statistical analysis been performed appropriately and rigorously? 

Reviewer #1: Yes

Reviewer #2: Yes

3. Have the authors made all data underlying the findings in their manuscript fully available?

Reviewer #1: Yes

Reviewer #2: Yes

4. Is the manuscript presented in an intelligible fashion and written in standard English?

Reviewer #1: Yes

Reviewer #2: Yes

5. Review Comments to the Author

Reviewer #1: General comments:

Current manuscript is an appropriately structured meta-analysis with 35 pages; 2 tables; 4 figures and 24 references. In general, the manuscript is well-written, but there is some grammar unpunctuality. The topic of this meta-analysis is interesting and have a great clinical relevance, however numerous previous meta-analyses were conducted on the topic. The novelty of the findings of current study is based on the results of one recently published big multinational mRCT (MYRIAD).

Specific revision comments:

-Major

ABSTRACT

- Few grammar inaccuracies and unpunctuality

METHODS

- Postoperative safety outcomes were collected on the longest follow-up available. Perhaps, it can add an additional bias, because adverse events occurring after several months are not necessarily due to the previous anaesthetic technique.

- Trial Sequential Analysis (TSA) would improve the value of the study.

RESULTS

- Most of the included RCTs are small and therefore, none of the patients died in the operative period. Basically, three study gives the main result and one study have a weight of 38.1%, which is quite high compared to the number of included studies.

- In 1-year mortality one study gives more than 50% of the results.

- Number of included studies in the final analyses can be confusing because only 45 and 5 RCTs were included in the main analyses.

DISCUSSION

- Discussion of the high heterogeneities are insufficient. Authors should describe the potential underlying causes.

Reviewer #2: The authors conducted a meta-analysis of randomized controlled clinical trials examining the efficacy of volatile anesthetics compared with total intravenous anesthesia to provide myocardial protection against ischemic injury in patients undergoing coronary artery surgery. A total of 90 trials involving 14,598 patients were included in the analysis. The results indicate that use of volatile anesthetics was not associated with improvements in outcome compared with total intravenous anesthesia. The findings are not surprising in the least. Giovanni Landoni and colleagues recently published a large prospective multicenter randomized clinical trial in which volatile anesthetics and total intravenous anesthesia were directly compared in patients undergoing coronary artery surgery; the study was stopped for futility after 50% of the proposed 10,000 patients were enrolled (NEJM, 2019). This definitive clinical trial effectively ended any further discussion about the so-called “cardioprotective” actions of volatile anesthetics that were demonstrated repeatedly in many laboratory studies during the past two decades. It is not clear to this reviewer why yet another meta-analysis is needed after this clinical trial, as the evidence provided by any meta-analysis is certainly not as strong as that provided by a well-conducted large scale randomized prospective clinical trial. The findings of the current meta-analysis are undoubtedly heavily influenced by the results of Landoni et al, as they should be. The current work is conducted appropriately using PRISMA guidelines and its data are interpreted correctly. The manuscript is fairly well written, although there are a few spelling and grammatical errors present that the authors should correct.

Specific Comments

P 3 L 68 and P 14 L 287: Several meta-analyses did not support this hypothesis (see references #5 and #7) for details.

P 3 L 74: Given the size and quality of the study of Landoni et al, did the authors think that their meta-analysis would somehow reach conclusions that truly different from the NEJM trial?

P 4 L 79: What is the hypothesis of the current work? This should be explicitly stated.

P 4 L 82: Please state that PRISMA guidelines were followed at this point of the text.

6. PLOS authors have the option to publish the peer review history of their article (what does this mean?). If published, this will include your full peer review and any attached files.

Reviewer #1: Yes: Nagy Ádám

Reviewer #2: No

---

## [Author Response · Author response to Decision Letter 0]

10 Oct 2019

This rebuttal letter responded to each point raised by the academic editor and reviewers, and was uploaded as a separate file labeling 'Response to Reviewers'. We specified the changes made to address each point, indicated their locations in the response (in brackets), and highlighted the changes in the file labeling 'Revised Manuscript with Track Changes' (in red font). This letter was divided into three parts: “Response to the Comments of Academic Editor”, “Response to the Comments of Reviewer #1”, and “Response to the Comments of Reviewer #2”. The details were as follows:

Response to the Comments of Academic Editor

1. The multicenter large randomized clinical trial by Giovanni Landoni and colleagues recently published on NEJM, stopped for futility after 50% of the proposed patients were enrolled, should stop any other discussion on this specific topic. Evidence provided by a randomized study is certainly stronger then the results of a meta-analysis, and the findings of the current meta-analysis are influenced by the results of Landoni et al.

Please put in evidence on your limitations these comments, and answer to the question on the review section.

Reply: Our meta-analysis was heavily influenced by the Landoni et al. RCT. This was an important limitation of our study, and we added this issue to our limitations section (Page 23, Line 459-462, in red font). Moreover, in order to answer to the question on the review section, we discussed the reasons why our meta-analysis is needed, and the differences between the results of our meta-analysis and the Landoni et al. RCT (Page 20-21, Line 396-409, in red font). The details were as follows:

“Landoni et al. RCT encompassed 5400 patients, which was much larger than any of the previous RCTs on this topic. This largest RCT failed to reveal any benefits of volatile anesthetics in CABG patients, which was quite contrary to the results of several previous meta-analyses. However, whether this RCT had enough statistical power to counter the results of previous meta-analyses and end further discussions about the benefit of volatile anesthetics was not known. Thus, we performed an updated and comprehensive meta-analysis on this topic. After pooling all the available evidence, the majority of our results were similar to the results of Landoni et al.; however, the results for length of stay in ICU and hospital were not consistent with the results of Landoni et al. Furthermore, our TSA results indicated that current evidence for mortality as well as several other postoperative safety outcomes was insufficient and inconclusive, and more RCTs would be required to further clarify this issue. Thus, although of large sample size, the Landoni et al. RCT is not convincing enough to end any further discussion on this topic.”

2. The grammar should be correct by an English native speaker.

Reply: The grammar has already been corrected by an English native speaker, and we provided the certificate of English editing in the online submission system.

3. In the manuscript N2O is included among the "volatiles". I think this is a procedural mistake. I suggest you to change the title or alternatively remove these studies.

Reply: This is our mistake. We have removed the studies using N2O or xenon as the anesthetic. After carefully checking the 90 included RCTs, one RCT using xenon as the anesthetic was removed, and none of the RCTs used N2O as the anesthetic. Thus, after removing, 89 RCTs were remained for our meta-analysis (Page 9, Line 177, 183-184, in red font). Consequently, the results for operative mortality, length of stay in ICU, length of stay in hospital, myocardial infarction, arrhythmia, delirium, and acute kidney injury were changed (Page 10-17, Line 202-204, 237-242, 249-254, 275-278, 294-297, 311-313, 329-332, Figs 2 and 4, Tables 1 and 2, in red font).

4. Reply: We have made changes to our financial disclosure, and included our updated statement in the cover letter. Our financial disclosure statement was updated as follows:

“This study was supported by Natural Science Foundation of China：Evidence based establishment of evaluation index system for pediatric rational drug use in China (No. 81373381), and National Science and Technology Major Project: Construction of pediatric new drug clinical evaluation technology platform (No. 2017ZX09304029). The funders had no role in study design, data collection and analysis, decision to publish, or preparation of the manuscript.” (in red font)

5. To enhance the reproducibility of your results, we recommend that if applicable you deposit your laboratory protocols in protocols.io, where a protocol can be assigned its own identifier (DOI) such that it can be cited independently in the future. For instructions see: http://journals.plos.org/plosone/s/submission-guidelines#loc-laboratory-protocols

Reply: We have deposited our protocol in protocols.io; the identifier (DOI) is dx.doi.org/10.17504/protocols.io.7y2hpye. Moreover, we added the DOI link to the Methods section of our revised manuscript (Page 5, Line 89-90, in red font). 

• A rebuttal letter that responds to each point raised by the academic editor and reviewer(s). This letter should be uploaded as separate file and labeled 'Response to Reviewers'.

• A marked-up copy of your manuscript that highlights changes made to the original version. This file should be uploaded as separate file and labeled 'Revised Manuscript with Track Changes'.

• An unmarked version of your revised paper without tracked changes. This file should be uploaded as separate file and labeled 'Manuscript'.

Reply: In accordance with the above requirements, we prepared and uploaded 'Response to Reviewers', 'Revised Manuscript with Track Changes', and 'Manuscript' separately.

7. Please ensure that your manuscript meets PLOS ONE's style requirements, including those for file naming. The PLOS ONE style templates can be found at http://www.journals.plos.org/plosone/s/file?id=wjVg/PLOSOne_formatting_sample_main_body.pdf and http://www.journals.plos.org/plosone/s/file?id=ba62/PLOSOne_formatting_sample_title_authors_affiliations.pdf

Reply: We have checked our manuscript carefully on the basis of the PLOS ONE style templates, and tried our best to ensure that our manuscript met PLOS ONE's style requirements.

We apologize that this was not previously requested and look forward to receiving your revised manuscript,

Reply: We removed the funding information from our manuscript. Moreover, in the online submission system, we updated our Funding Statement as follows:

“This study was supported by Natural Science Foundation of China：Evidence based establishment of evaluation index system for pediatric rational drug use in China (No. 81373381), and National Science and Technology Major Project: Construction of pediatric new drug clinical evaluation technology platform (No. 2017ZX09304029). The funders had no role in study design, data collection and analysis, decision to publish, or preparation of the manuscript.”

Response to the Comments of Reviewer #1 

1. ABSTRACT - Few grammar inaccuracies and unpunctuality

Reply: The grammar has been corrected by an English native speaker, and we provided the certificate of English editing in the online submission system.

2. METHODS - Postoperative safety outcomes were collected on the longest follow-up available. Perhaps, it can add an additional bias, because adverse events occurring after several months are not necessarily due to the previous anaesthetic technique.

Reply: In order to reduce this bias, we changed our Methods as: “Postoperative safety outcomes occurring during hospitalization or within 30 days of operation were extracted for meta-analysis.” (Page 6, Line 129-130, in red font). Consequently, in our conventional meta-analysis, the results for arrhythmia, postoperative cognitive impairment, and the use of other mechanical circulatory support were changed (Page 13-18, Line 265, 271, 294-297, 318-321, 347-350, Tables 1 and 2, in red font).

3. METHODS - Trial Sequential Analysis (TSA) would improve the value of the study.

Reply: We performed trial sequential analysis (TSA) to determine whether the currently available evidence was sufficient and conclusive (Page 4, Line 82-84, in red font). The methods of TSA could be seen on Page 8, Line 162-173 (in red font), and the results of TSA could be seen on Page 10-12, 15-18, Line 205-208, 223-226, 243-246, 255-256, 279-281, 288-291, 298-299, 305-308, 314-315, 324-326, 333-335, 341-344, 351-353 (in red font), Figs 3 and 5, S2-S12 Figs. In summary, TSA showed that the results for length of stay in hospital, myocardial infarction, arrhythmia, delirium, postoperative cognitive impairment, acute kidney injury, and the use of other mechanical circulatory support were conclusive, while the results for operative mortality, one-year mortality, length of stay in ICU, heart failure, stroke, and the use of IABP were inconclusive (Page 19, Line 370-375, in red font). Moreover, when considering the TSA results, the quality of evidence for our outcomes was also changed (Page 10-11, 15-16, 18, 22, Line 209, 227, 292, 309, 345, 428-439, S5 Table, in red font).

4. RESULTS - Most of the included RCTs are small and therefore, none of the patients died in the operative period. Basically, three study gives the main result and one study have a weight of 38.1%, which is quite high compared to the number of included studies. In 1-year mortality one study gives more than 50% of the results.

Reply: Our meta-analysis was heavily influenced by three large-scale RCTs, especially the Landoni et al. RCT. This was an important limitation of our study, and thus we added this issue to our limitations section (Page 23, Line 459-462, in red font). Moreover, we also discussed the differences between the results of our meta-analysis and the Landoni et al. RCT, and the reasons why our meta-analysis is needed (Page 20-21, Line 396-409, in red font).

5. RESULTS - Number of included studies in the final analyses can be confusing because only 44 and 5 RCTs were included in the main analyses.

Reply: We included RCTs that reported at least one of our outcomes (primary outcomes or secondary outcomes). According to this criteria, a total of 89 RCTs were included. For primary outcomes, 44 RCTs were included for operative mortality, and 5 RCTs were included for one-year mortality. While the remaining RCTs only reported the secondary outcomes.

6. DISCUSSION - Discussion of the high heterogeneities are insufficient. Authors should describe the potential underlying causes.

Reply: We described the potential underlying causes on Page 21, Line 412-419 (in red font). One possible reason was that some of the included RCTs provided the median, range, and/or the first and third quartiles for these outcomes. Transformation of these data to sample means and SDs for meta-analysis may have introduced errors, thus causing heterogeneity between studies [1]. Other reasons may include variations in ethnicity, age, severity of disease, treatment regimens, operation types and anesthesia schemes. It is noteworthy that the heterogeneities may also be attributed to variations in settings, hospitals, medical insurance systems, economy levels and cultural differences.

References: 1. Wan X, Wang W, Liu J, Tong T. Estimating the sample mean and standard deviation from the sample size, median, range and/or interquartile range. BMC Med Res Methodol. 2014;14:135.

Response to the Comments of Reviewer #2 

1. Giovanni Landoni and colleagues recently published a large prospective multicenter randomized clinical trial in which volatile anesthetics and total intravenous anesthesia were directly compared in patients undergoing coronary artery surgery; the study was stopped for futility after 50% of the proposed 10,000 patients were enrolled (NEJM, 2019). This definitive clinical trial effectively ended any further discussion about the so-called “cardioprotective” actions of volatile anesthetics that were demonstrated repeatedly in many laboratory studies during the past two decades. It is not clear to this reviewer why yet another meta-analysis is needed after this clinical trial, as the evidence provided by any meta-analysis is certainly not as strong as that provided by a well-conducted large scale randomized prospective clinical trial.

Given the size and quality of the study of Landoni et al, did the authors think that their meta-analysis would somehow reach conclusions that truly different from the NEJM trial?

Reply: In the Discussion section, we discussed the reasons why our meta-analysis is needed, and the differences between the results of our meta-analysis and the Landoni et al. RCT (Page 20-21, Line 396-409, in red font). The details were as follows:

“Landoni et al. RCT encompassed 5400 patients, which was much larger than any of the previous RCTs on this topic. This largest RCT failed to reveal any benefits of volatile anesthetics in CABG patients, which was quite contrary to the results of several previous meta-analyses. However, whether this RCT had enough statistical power to counter the results of previous meta-analyses and end further discussions about the benefit of volatile anesthetics was not known. Thus, we performed an updated and comprehensive meta-analysis on this topic. After pooling all the available evidence, the majority of our results were similar to the results of Landoni et al.; however, the results for length of stay in ICU and hospital were not consistent with the results of Landoni et al. Furthermore, our TSA results indicated that current evidence for mortality as well as several other postoperative safety outcomes was insufficient and inconclusive, and more RCTs would be required to further clarify this issue. Thus, although of large sample size, the Landoni et al. RCT is not convincing enough to end any further discussion on this topic.”

2. There are a few spelling and grammatical errors present that the authors should correct.

Reply: The spelling and grammatical errors have been corrected by an English native speaker, and we provided the certificate of English editing in the online submission system.

3. P 3 L 68 and P 14 L 287: Several meta-analyses did not support this hypothesis (see references #5 and #7) for details.

Reply: It is our mistake. We carefully checked the previous published meta-analyses and found that two meta-analyses2,3 published in 2006 did not support this hypothesis. Thus, we modified P 3 L 68 as follows: 

“However, the benefits of volatile anesthetics in CABG patients is an intensely disputed topic. In 2006, two meta-analyses of randomized controlled trials (RCTs) suggested that volatile anesthetics were not associated with reduced mortality when compared with total intravenous anesthesia (TIVA) in CABG surgery [2,3]. On the other hand, meta-analyses published later have shown a reduced mortality associated with volatile anesthetics in CABG surgeries.” (Page 3-4, Line 65-70, in red font)

Moreover, we modified P 14 L 287 as follows:

“Our meta-analysis did not reveal reductions in mortality by the use of volatile anesthetics in CABG surgery, a result which was similar to those of meta-analyses published in 2006. However, the meta-analyses published after 2006 were not consistent with our results.” (Page 19, Line 377-380, in red font)

References:

2. Yu CH, Beattie WS. The effects of volatile anesthetics on cardiac ischemic complications and mortality in CABG: a meta-analysis. Canadian journal of anaesthesia = Journal canadien d'anesthesie. 2006;53(9):906-918. 

3. Symons JA, Myles PS. Myocardial protection with volatile anaesthetic agents during coronary artery bypass surgery: A meta-analysis. British journal of anaesthesia. 2006;97(2):127-136.

4. P 4 L 79: What is the hypothesis of the current work? This should be explicitly stated.

Reply: Our hypothesis was that the use of volatile anesthetics during CABG does not result in lower mortality than the use of total intravenous anesthesia (TIVA). This hypothesis was explicitly stated on Page 4, Line 78-80 (in red font). 

5. P 4 L 82: Please state that PRISMA guidelines were followed at this point of the text.

Reply: We stated as: “This meta-analysis was conducted in accordance with the Cochrane Handbook as well as the Preferred Reporting Items for Systematic Reviews and Meta-Analyses (PRISMA) guidelines.” (Page 4-5, Line 87-89, in red font)

---

## [Editor Report · Decision Letter 1]

17 Oct 2019

Volatile anesthetics versus total intravenous anesthesia in patients undergoing coronary artery bypass grafting: An updated meta-analysis and trial sequential analysis of randomized controlled trials

PONE-D-19-23473R1

Dear Dr. Zhang,

We are pleased to inform you that your manuscript has been judged scientifically suitable for publication and will be formally accepted for publication once it complies with all outstanding technical requirements.

With kind regards,

Martina Crivellari

Academic Editor

PLOS ONE
---

## [Editor Report · Acceptance letter]

21 Oct 2019

PONE-D-19-23473R1 

Volatile anesthetics versus total intravenous anesthesia in patients undergoing coronary artery bypass grafting: An updated meta-analysis and trial sequential analysis of randomized controlled trials 

Dear Dr. Zhang:

I am pleased to inform you that your manuscript has been deemed suitable for publication in PLOS ONE. Congratulations! Your manuscript is now with our production department. 

With kind regards,

on behalf of

Dr. Martina Crivellari 

Academic Editor

PLOS ONE